# Cuscutae Japonicae Semen Ameliorates Memory Dysfunction by Rescuing Synaptic Damage in Alzheimer’s Disease Models

**DOI:** 10.3390/nu11112591

**Published:** 2019-10-28

**Authors:** In Gyoung Ju, Namkwon Kim, Jin Gyu Choi, Jong Kil Lee, Myung Sook Oh

**Affiliations:** 1Department of Life and Nanopharmaceutical Sciences, Graduate school, Kyung Hee University, 26 Kyungheedae-ro, Dongdaemun-gu, Seoul 02447, Korea; igju801@khu.ac.kr (I.G.J.); kop03@khu.ac.kr (N.K.); 2BK21 PLUS Integrated Education and Research Center for Nature-inspired Drug Development Targeting Healthy Aging, Kyung Hee University, 26 Kyungheedae-ro, Dongdaemun-gu, Seoul 02447, Korea; choijg2002@khu.ac.kr; 3Department of Pharmacy, College of Pharmacy, Kyung Hee University, 26 Kyungheedae-ro, Dongdaemun-gu, Seoul 02447, Korea; jklee3984@khu.ac.kr; 4Department of Oriental Pharmaceutical Science, College of Pharmacy and Kyung Hee East-West Pharmaceutical Research Institute, Kyung Hee University, 26 Kyungheedae-ro, Dongdaemun-gu, Seoul 02447, Korea

**Keywords:** Alzheimer’s disease, Cuscutae Japonicae Semen, synaptic damage, caspase-3

## Abstract

Alzheimer’s disease (AD) is the most common type of dementia in the elderly. It is characterized by the accumulation of amyloid-beta (Aβ) and progressive cognitive impairment. To alleviate the symptoms of AD, functional foods and nutrients have been used for centuries. In this study, we investigated whether Cuscutae Japonicae Semen (CJS), a medicinal food traditionally used in East Asia, has effects on memory improvement and synapse protection in AD. We orally administered CJS to 5x familiar AD (5xFAD) transgenic mice and performed the Morris water maze test. The results showed that CJS treatment ameliorated the decline of memory function. Then, we demonstrated that CJS attenuated the degeneration of pre- and post-synaptic proteins in the hippocampi of 5xFAD mice. To demonstrate the effects of CJS in vitro, we treated Aβ in primary neuronal culture with CJS and observed that CJS rescued the loss of functional synapses. The protective effects of CJS on the synapse were due to the inhibition of activated caspase-3 expression. Additionally, CJS inhibited the phosphorylation of glycogen synthase kinase-3β and tau proteins, which contribute to synaptic dysfunction. Taken together, our results suggest that CJS is efficient in alleviating memory loss by rescuing caspase-3-mediated synaptic damage in AD treatment.

## 1. Introduction

Alzheimer’s disease (AD), the most common type of dementia, is a progressive and irreversible neurodegenerative disease [1]. The major symptoms of AD include memory loss, cognitive decline, and personality change [2]. AD is histologically characterized by the shrinkage of the cortex and hippocampus, the accumulation of amyloid plaques and tau tangles, and nerve cell damage [3]. Although the etiology of AD is not clearly elucidated, excessive accumulation of amyloid-beta (Aβ) is considered to be a major pathological cause of AD that results in oxidative stress, inflammation, excitotoxicity, tau pathology, and apoptosis [4,5]. Prolonged hyperexcitability, abnormal tau physiology, and localized apoptosis in synaptic regions induce synaptic dysfunction, which directly correlates to the memory decline associated with AD [6,7,8,9]. In the pathway of Aβ-mediated synaptic dysfunction, caspase-3, localized in the synaptic region, is known to play a pivotal role by inducing synaptic elimination and abnormal tau physiology, which lead to synaptic dysfunction [10].

Several studies have reported that various foods and nutrients exert preventive effects on AD. For example, ginger is known to have pharmacotherapeutic potential in neurological disorders by protecting neurons against Aβ neurotoxicity, modulating cholinergic signaling, and enhancing memory [11,12]. Additionally, a number of studies showed that turmeric curry (*Curcuma longa* Linné) attenuates neurotoxic and behavioral damage in various AD models by inhibiting Aβ accumulation and protecting neurons from oxidative damage, apoptosis, and hyperexcitation, suggesting that it could reverse AD pathology [13,14]. Moreover, Danshen (*Salvia miltorrhiza* Bunge) has been demonstrated to exert neuroprotective effects by attenuating mitochondria-dependent apoptosis and inducing the neuronal differentiation, indicating its potential as a treatment for AD [15,16].

Cuscutae Japonicae Semen (CJS) is a dried seed of *Cuscuta japonica* Choisy (Convolvulaceae), also known as “Japanese dodder”, which is widely distributed in East Asia and America [17,18]. CJS has been consumed as a medicinal food for the treatment of glycosuria and impotence, as it nourishes liver function and strengthens the kidneys [19,20,21]. A variety of studies reported its pharmacological activities including inhibitory activities on angiotensin-converting enzymes, protective effects on human sperm, and suppressive effects on melanin synthesis [18,20,21]. Notably, we reported that CJS exerts memory-enhancing effects, thereby suggesting that CJS may have potential as a functional food that improve memory function in various neurodegenerative diseases [22]. However, the effects of CJS on AD pathology are still unknown. Therefore, the aims of this study were to investigate whether CJS attenuates memory impairment and synaptic damage caused by Aβ in 5x familiar AD (5xFAD) transgenic mice, and to find out its mode of action in primary cultured hippocampal neurons damaged by Aβ.

## 2. Materials and Methods

### 2.1. Materials

Neurobasal media (NM) and B27 were purchased from Gibco Industries Inc. (Auckland, NZ). Penicillin-streptomycin was purchased from Hyclone Laboratories, Inc. (Logan, UT, USA). Mouse anti-glycogen synthase kinase-3β (GSK-3β), goat anti-phosphorylated GSK-3β (ser9), mouse anti-β-actin, and anti-goat horseradish peroxidase (HRP) secondary antibody were purchased from Santa Cruz Biotechnology (Temecula, CA, USA). Rabbit anti-post-synaptic density protein 95 (PSD-95) was purchased from Abcam (Cambridge, UK). Rabbit anti-phosphorylated-tau (ser262) was purchased from Thermo Fisher Scientific Inc. (Waltham, MA, USA). Rabbit anti-cleaved-caspase-3 and rabbit anti-tau were purchased from Cell Signaling Technology (Danvers, MA, USA). Skim milk was purchased from BD Transduction Laboratories (Franklin Lakes, NJ, USA). Polyvinylidene difluoride (PVDF) was purchased from Millipore (Burlington, USA). Horse anti-mouse DyLight 488, goat anti-rabbit DyLight 594, and 4′,6-diamidino-2-phenylindole (DAPI) were purchased from Vector Laboratories (Burlingame, CA, USA). Anti-rabbit and anti-mouse HRP secondary antibodies were purchased from Enzo Life Science Inc. (Farmingdale, NY, USA). Tetramethylethylenediamine, protein assay reagent, acrylamide, and enhanced chemiluminescence (ECL) reagent were purchased from Bio-Rad Laboratories (Hercules, CA, USA). PRO-PREP™ Protein extraction solution was purchased from iNtRON biotechnology (Gyeonggi, Republic of Korea). Fluorescent mounting medium was purchased from Dako Cytomation (Glostrup, Denmark). Mouse anti-synaptophysin (SYP) and the other reagents were purchased from Sigma-Aldrich (St. Louis, MO, USA) unless noted. Aβ_25–35_ was purchased from AnaSpec (Fremont, CA, USA) and reconstituted in sterile water at a concentration of 500 μM and incubated at 37 °C for 72 hours to form amyloid plaque as previously described [23]. 

### 2.2. Preparation of the CJS Extract (CJSE)

CJSE was prepared by the same procedure as in a previous study [22]. Briefly, dried CJS was purchased from Cheon-ji-ga Herb (Seoul, Republic of Korea) and authenticated by Professor Ho-Young Choi at College of Korean Medicine, Kyung Hee University (Seoul, Republic of Korea). The voucher specimen (BON18121201) was deposited in the herbarium of College of Pharmacy, Kyung Hee University (Seoul, Republic of Korea). CJS was extracted with sonication in distilled water three times for 1 hour. Then, the extract was filtered, evaporated in a rotary vacuum evaporator, and finally lyophilized. The powder (yield, 20.53%) was kept at −20 °C.

### 2.3. Animals and Administration

5xFAD mice were purchased from The Jackson Laboratory (Bar Harbor, ME, USA). 5xFAD mutations include APP KM670/671NL (Swedish), APPI716V (Florida), APPV717I (London), PSEN1 M146L, and PSEN1 L286V, resulting in early and aggressive Aβ accumulation related to memory deficits [24]. For the experiments, we used female wild-type (WT) and 5xFAD mice. Mice were divided into three groups (WT, 5xFAD, and 5xFAD+CJSE; *n* = 5 per group; sample sizes were determined with adequate statistical power on the basis of the literature and previous experience) and housed at a constant temperature (23 ± 1 °C), humidity (60 ± 10%), and a 12 hours light/dark cycle. Animals had free access to food and water. CJSE at 100 mg/kg was orally administered using a Zonde needle every 6 days per week, at from 20 to 28 weeks of age, referring to the dose determined in the previous study [22]. All animal studies were performed in accordance with the “Guide for the Care and Use of Laboratory Animals, 8th edition” (National Institutes of Health, 2011) and approved by the “Animal Care and Use Guidelines” of Kyung Hee University, Seoul, Republic of Korea (the approval number: KHUASP(SE)-17-154).

### 2.4. Morris Water Maze Test

The Morris water maze (MWM) test was conducted in a dark room. The maze used for the test was a round tank (height of 45 cm and diameter of 90 cm) with four different visual cues on the wall. One hidden platform (height of 29 cm and diameter of 10 cm) was placed in the center of a quadrant in the tank, which was conceptually divided. We filled the tank with water (24 ± 1 °C) up to 1 cm above the hidden platform and added white opaque nontoxic paint to the water to hinder visibility. All mice were habituated to the maze one day before the trials, and were subjected to two trials per day for 5 days. In each of the two trials, animals were randomly placed at different starting positions, equally spaced around the perimeter of the pool, and given 60 seconds to find the submerged platform. If a mouse did not mount the platform within 60 seconds, it was guided to the platform. The time to mount the platform was recorded as the latency for each trial. All mice were allowed to remain on the platform for 10 seconds before being returned to a cage. 

### 2.5. Brain Tissue Preparation

The mice were anesthetized and rapidly perfused transcardially with 0.05 M phosphate-buffered saline (PBS), followed by 4% paraformaldehyde (PFA) in 0.1 M phosphate buffer. Brains were taken out, post-fixed overnight in 4% PFA, and immersed in a solution containing 30% sucrose in PBS for cryoprotection. Serial 25 μm-thick coronal sections were cut on a freezing sliding microtome (Leica Microsystems Inc., Nussloch, Germany) and stored in cryoprotectant (25% ethylene glycol, 25% glycerol, and 0.05 M phosphate buffer) at 4 °C until use.

### 2.6. Immunohistochemistry

For the immunohistochemical study, hippocampal sections were selected according to the mouse brain atlas; from −1.94 to −2.30 mm following coordinates from the bregma [25]. Free-floating brain sections were pre-treated with 1% hydrogen peroxide in PBS for 15 minutes to remove endogenous peroxidase activity and incubated with anti-PSD-95 or anti-SYP antibody overnight at 4 °C in the presence of 0.3% Triton X-100. After rinsing in PBS, they were then incubated with biotinylated anti-rabbit or anti-mouse immunoglobulin G for 1 hour and with avidin–biotin complex mixture for 1 hour at room temperature. Peroxidase activity was visualized by incubating sections with 3,3′-diaminobenzidine in 0.05 M tris-buffer. After several rinses with PBS, sections were mounted on gelatin-coated slides and cover-slipped using DPX mountant for histology. The images were photographed using an optical bright-field microscope (Olympus Microscope System BX51; Olympus, Tokyo, Japan).

### 2.7. Primary Cultures of Hippocampal Neuronal Cells

Hippocampal neuronal cultures were prepared from the hippocampus of 18-day-old embryos of pregnant ICR mice (Daehan Biolink Co. Ltd., Eumseong, Republic of Korea). Hippocampi were dissected, collected, dissociated, and plated in poly-L-lysine pre-coated 24-well plates with coverslips and 60-mm dishes at densities of 3.0 × 10^4^ cells/well and 1.0 × 10^6^ cells/dish, respectively. Cultures were maintained in a humidified incubator of 5% CO_2_ at 37 °C in a NM with 2 mM glutamine, 2% B27, and 1% penicillin/streptomycin. The medium was replaced with a new medium once every 3 days. On day in vitro 7 (DIV7), cells were treated with CJSE at 10 μg/mL for 24 hours. The treated cells were incubated with 8 μM Aβ_25–35_ for the last 23 hours. An equal volume of vehicle was given to the normal and Aβ groups.

### 2.8. Western Blotting

Western blotting was performed according to the method described previously [26]. The primary cultured cells were lysed with protein lysis buffer (*n* = 3 or 4 per group). Cell lysates were separated by sodium dodecyl sulfate-polyacrylamide gel electrophoresis and transferred to PVDF membranes. The membranes were incubated with 5% skim milk for 1 hour and with primary antibodies for overnight at 4 °C. Then, the membranes were incubated with HRP-conjugated secondary antibodies for 1  h. Immunoreactive bands were detected using an ECL reagent. The visualization and quantification of band intensity were performed using Image Lab Software (Bio-Rad, Hercules, CA, USA).

### 2.9. Immunocytochemistry and Synaptic Density Analysis

Mouse hippocampal cells on cover slips were fixed with 4% PFA at room temperature for 30 minutes (*n* = 5 per group). Then, the cells were rinsed with PBS and incubated with anti-SYP and anti-PSD-95 antibodies overnight at 4 °C in the presence of 0.3% Triton X-100. After rinsing in PBS, cells were incubated with anti-mouse DyLight 488 and anti-rabbit DyLight 594 for 70 minutes and then with DAPI for 20 minutes at room temperature. The cover slips were mounted on slides. Images of neurons stained with SYP and PSD-95 were obtained using a K1-Fluo confocal microscope (Nanoscope Systems, Daejeon, Republic of Korea). Since only the co-localization of pre-synaptic and post-synaptic markers represent active synapses, we assessed synaptic density by analyzing synaptic puncta containing SYP and PSD-95 with an ImageJ plugin, SynapCountJ (http://imagejdocu.tudor.lu/doku.php?id=plugin:utilities:synapsescountj:start), as previously described [27]. SynapCountJ semi-automatically quantified the number of synapses in the neuron cultures, and synaptic density was calculated as the density of synapses per 100 μm. For analysis, over 20 dendrites were used three or more times in independent cultures. Dendrites and regions of interest were randomly chosen, and the threshold of intensity was fixed during the analysis.

### 2.10. Statistical Analysis

All statistical parameters were calculated using GraphPad Prism 5.0 software (GraphPad Software, San Diego, CA, USA). Values were expressed as the mean ± standard error of the mean (S.E.M.) and analyzed using either two-way analysis of variance (ANOVA) followed by Bonferroni post-hoc test or an unpaired Student’s t-test. Differences with a p-value less than 0.05 were considered statistically significant. 

## 3. Results

### 3.1. CJSE Attenuated Memory Impairment in 5xFAD Mice

To investigate whether CJSE administration affected behavioral change in the AD mice models, we administered CJSE into 5xFAD mice whose genes are mutated to generate excessive Aβ accumulation and represent the pathological features of familiar AD. After administration for 8 weeks, we performed the MWM test, which is a widely accepted method to measure the capacity of spatial learning and memory of mice [28]. Figure 1A summarizes the experimental time scheme. We measured the escape latencies to find the hidden platform for two trials per day. We observed that the escape latencies of the 5xFAD mice were significantly longer than those of WT mice on the last day. The escape latencies of the CJSE-treated 5xFAD mice were significantly shorter than those of the 5xFAD mice on the fourth and last day (Figure 1B). 

### 3.2. CJSE Rescued Synaptic Loss in 5xFAD Mice

To find out how CJSE ameliorated memory impairment, we investigated whether CJSE inhibits synaptic loss, which correlates primarily with the memory deficits of AD [7]. We measured the protein levels of SYP and PSD-95, which are pre- and post-synaptic markers, respectively, in the hippocampus. We observed that the optical densities of SYP in the cornu ammonis 3 (CA3) and molecular layer of the dentate gyrus (DG-ML) regions were lower in the 5xFAD group than in the WT group. However, in the CJSE-treated 5xFAD group, the optical densities were significantly higher than those in the 5xFAD group (Figure 2A–C). Similarly, we calculated the optical densities of PSD-95 in the same regions of hippocampus, and observed that optical densities in the 5xFAD mice were significantly reduced compared to WT mice, and CJSE treatment attenuated the reduction (Figure 2D–F).

### 3.3. CJSE Protected Synaptic Loss Induced by Aβ in Mouse Primary Hippocampal Neurons

Our in vivo results indicated that CJSE rescued synaptic loss against Aβ-induced synaptotoxicity, resulting in the attenuation of memory decline. To demonstrate the effects of CJSE in vitro, we assessed the protective effects of CJSE against Aβ-induced synaptotoxicity in hippocampal neurons cultured from mouse embryos. We measured the protein expression levels of SYP and PSD-95 by Western blotting. We observed that the protein levels of SYP and PSD-95 decreased after treatment with Aβ plaque. However, CJSE treatment significantly inhibited the decrease in synaptic protein levels compared to those in the Aβ-treated group (Figure 3A–C). In addition, since the co-localization of SYP and PSD-95 indicates functional synapse density, we measured synaptic density by calculating the percent of merged puncta of SYP and PSD-95 using immunocytochemical analysis [29]. We observed that the treatment of Aβ plaque decreased synaptic density, as shown by reduced PSD-95/SYP co-localized puncta. However, CJSE increased the amount of co-localized puncta, indicating the inhibition of the decrease in functional synapses (Figure 3D,E). These results verify that CJSE protects the synapse from Aβ-induced synaptotoxicity, supporting the in vivo data described above.

### 3.4. CJSE Suppressed the Activation of Caspase-3

There is emerging evidence of the non-apoptotic roles of caspase-3 in the synaptic terminal [30]. Caspase-3 is activated by cleavage after stimulation by Aβ toxicity, and then plays a pivotal role in synaptic elimination by inducing synaptic apoptosis [31]. In this study, to demonstrate how CJSE attenuated the degeneration of the synapse induced by Aβ toxicity, we explored the effects of CJSE on caspase-3 by measuring the protein levels of activated caspase-3. We observed that Aβ elevated the levels of cleaved caspase-3 compared to levels in the normal group. However, CJSE treatment attenuated the increase (Figure 4). These results were also confirmed in vivo, as observing that cleaved caspase-3-immunoreactivity was more intense in the hippocampal CA3 region of 5xFAD mice than that in WT mice, whereas that was reduced in CJSE-treated 5xFAD mice (Appendix A).

### 3.5. CJSE Regulated GSK-3β Phosphorylation and Tau Phosphorylation

Caspase-3 induces the activation of GSK-3β and the cleavage of tau, which contribute to synaptic dysfunction. GSK-3β becomes activated by dephosphorylation and then regulates synaptic plasticity [32]. Additionally, GSK-3β is a serine/threonine kinase of tau protein. Cleaved and phosphorylated tau is reported to mediate Aβ-induced synaptotoxicity by reducing dendritic spine density [33]. In this study, we measured the ratio of phosphorylated (inactive) GSK-3β and observed that the ratio was decreased by Aβ treatment compared to the normal group. However, CJSE significantly increased the ratio (Figure 5A,B). Additionally, we demonstrated that the phosphorylation of tau increased in the Aβ-only treated group compared to the normal group, whereas in the CJSE-treated group, the phosphorylation of tau was significantly reduced (Figure 5C,D).

## 4. Discussion

In this study, we assessed the ameliorative effects of CJS on Aβ-induced memory impairment and synaptotoxicity in 5xFAD mice and mouse primary neuronal culture. We revealed that CJS prevented memory dysfunction and inhibited the loss of synaptic proteins by suppressing caspase-3 activation. 

We performed the MWM test to assess the behavioral change related to memory function. We observed that the escape time of CJS-treated 5xFAD mice was reduced in a time-dependent manner. We measured the Aβ deposits by thioflavin S staining and immunostaining with Aβ_1-16_ antibody to examine whether memory improvement of CJSE treatment was associated with the Aβ toxicity. However, CJSE did not exhibit any effects on Aβ deposits compared to the 5xFAD mice (Appendix A), indicating that the memory improvement induced by CJS treatment is not due to the effects on Aβ accumulation. Since synaptic failure is considered to be the major neurobiological mechanism of memory decline in AD, we assessed the effects of CJS on synaptic loss by immunostaining synaptic proteins [34]. Reportedly, synaptic degeneration presents in 5xFAD mice at from 5 to 9 months of age [35,36,37]. In the present study, we observed the synaptic loss in 28-week-old 5xFAD mice and found that CJS protects synaptic loss induced by Aβ. These results indicate that CJS does not inhibit Aβ accumulation, but attenuates memory impairment by preventing synaptic loss that occurs after excessive accumulation of Aβ.

Aβ contributes to the elimination of the synapse by inducing the activation of caspase-3 localized in synaptic terminals, which leads to synaptic apoptosis [38]. It was reported that caspase-3 is preferentially localized in post-synaptic density regions and increased in AD patients, implying a pathological roles of caspase-3 in the synaptic dysfunction expressed in AD [39]. Liu et al. revealed that Aβ induces synaptic degeneration which mediates caspase-8 and caspase-3 activation after the suppression of the N-methyl-D-aspartate (NMDA) receptor 2A and the activation of NMDA receptor 2B [31]. Additionally, Jang et al. demonstrated that Aβ oligomers induce the loss of pre-synaptic proteins in primary neuronal culture in a caspase-dependent manner [40]. In this study, we suggested that CJS can protect synapses from being eliminated by showing that CJS inhibits the expression of cleaved caspase-3 in primary hippocampal cells.

In addition, caspase is involved in the disruption of synaptic function. Caspase-3 has been demonstrated to cleave protein kinase B, resulting in the dephosphorylation of GSK-3β and the phosphorylation of tau protein, which abolish synaptic function [41]. Reportedly, GSK-3β plays a pivotal role in synaptic plasticity by inducing long-term depression [10,32]. Additionally, GSK-3β is a serine/threonine kinase that phosphorylates various sites of tau protein [9]. Phosphorylated tau is also regarded as a major protein in the disruption of synaptic transmission by modulating synaptic vesicle release, stimulating neurotransmitter receptors, and interacting with PSD-95 [33,42]. Furthermore, phosphorylated tau becomes truncated, misfolded, and aggregated to form oligomers and fibrils, and then induces cytoskeletal dysfunction and synaptic degeneration [3]. Our results demonstrated the inhibitory effects of CJS on GSK-3β activation and tau phosphorylation, suggesting the potential of CJS as a compelling agent to reduce synaptic damage caused by Aβ.

CJS is rich in polysaccharides, as it contains approximately 5% polysaccharide content [43]. Additionally, it is reported that CJS contains γ- and δ-tocopherol, which are well-known antioxidants [44]. CJS also contains secondary metabolites such as hyperoside, 3,5-di-O-caffeoylquinic acid, methyl 3,5-di-O-caffeoylquinate, 3,4-di-O-caffeoylquinic acid, and methyl 3,4-di-O-caffeoylquinate [18,22]. Hyperoside is reported to exert neuroprotective effects against Aβ toxicity by the inhibition of apoptosis [45]. Moreover, various types of caffeoylquinic acids have been studied to have protective effects on the aggregation and toxicity of Aβ [46]. Remarkably, 3,5-di-O-caffeoylquinic acid is reported to possess neuroprotective effects by up-regulating phosphoglycerate kinase-1 in human neuroblastoma SH-SY5Y cells and senescence-accelerated prone mice [47]. Based on these previous findings, we can conjecture that these components may contribute to the protective effects of CJS against Aβ toxicity. However, a precise analysis of the nutrients of CJS has not been performed. Therefore, studies on the neuroactive nutrients involved in CJS are needed. 

Our previous study demonstrated that CJS enhanced memory function by inducing adult neurogenesis [22]. These results suggested that CJS is a potential candidate as a therapeutic for various neurodegenerative disorders, including AD and other types of dementia. In this study, we demonstrated that CJS attenuated the memory decline in AD by the reduction of synaptic damage caused by Aβ. Thus, based on these two independent studies, we expect that CJS may be used as a preventive and therapeutic supplement for AD patients who have memory dysfunction.

## Figures and Tables

**Figure 1 nutrients-11-02591-f001:**
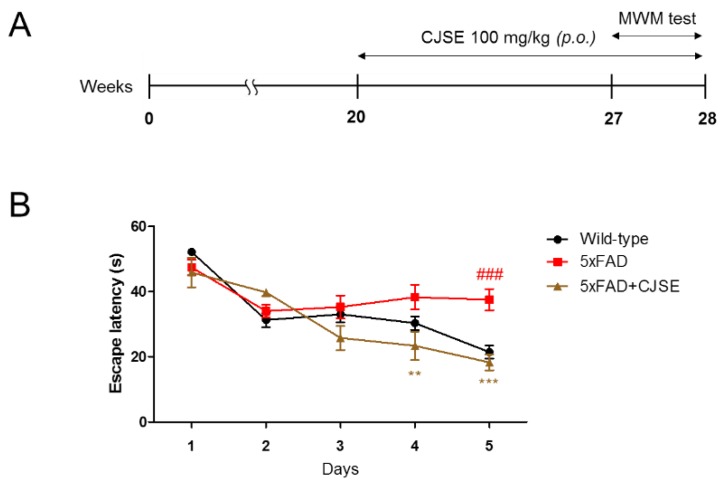
Effects of Cuscutae Japonicae Semen extract (CSJE) treatment on memory impairment in 5x familiar AD (5xFAD) mice. (**A**) Time scheme of the CJSE treatment and Morris water maze (MWM) test. (**B**) Escape latencies of wild-type, 5xFAD, or CJSE-treated 5xFAD mice. Values are indicated as the mean ± S.E.M. ### *p* < 0.001 compared to wild-type mice; ** *p* < 0.01 and *** *p* < 0.001 compared to 5xFAD mice.

**Figure 2 nutrients-11-02591-f002:**
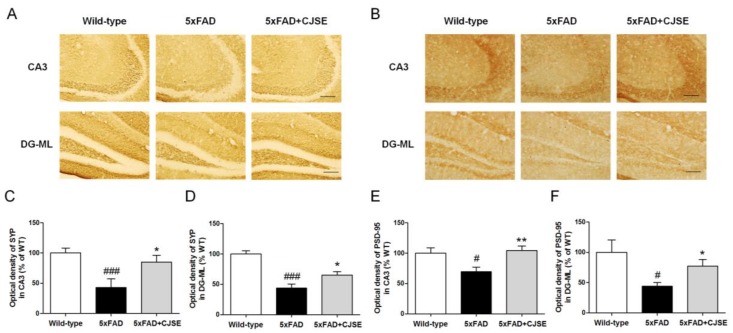
Effects of CJSE on synaptic loss in 5xFAD mice. (**A**,**B**) Representative photomicrographs of synaptophysin (SYP) (**A**) and post-synaptic density protein 95 (PSD-95) (**B**) immunoreactive regions in hippocampal CA3 and DG-ML; scale bar = 100 μm. (**C**,**D**) Quantifications of SYP immunoreactive regions in hippocampal CA3 (**C**) and DG-ML (**D**). (**E**,**F**) Quantifications of PSD-95 immunoreactive regions in hippocampal CA3 (**E**) and DG-ML (**F**). Values are indicated as the mean ± S.E.M. # *p* < 0.05 and ### *p* < 0.001 compared to wild-type mice; * *p* < 0.05 and ** *p* < 0.01 compared to 5xFAD mice. CA3; cornu ammonis 3, DG-ML; molecular layer of the dentate gyrus.

**Figure 3 nutrients-11-02591-f003:**
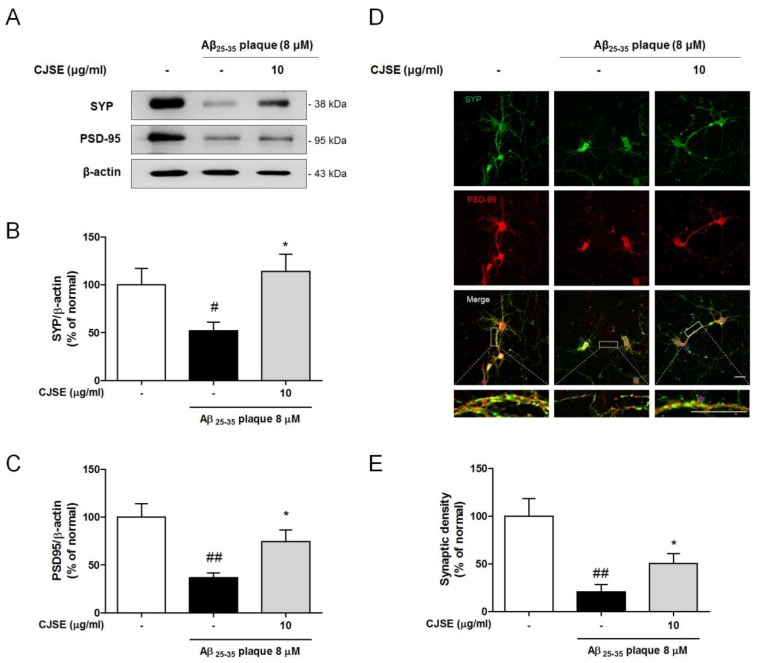
Effects of CJSE on loss of synaptic protein and functional synaptic density in mouse primary neuronal culture. (**A**) Representative band images of SYP and PSD-95 expression. (**B**,**C**) Quantifications of SYP (**B**) and PSD-95 (**C**) levels were normalized to β-actin. (**D**) Representative photomicrographs of SYP (green) and PSD-95 (red)-stained neurons. Bottom insets show higher magnification; scale bar = 100 μm. (**E**) Quantifications of synaptic density (co-localized puncta of SYP and PSD-95). Values are indicated as the mean ± S.E.M. # *p* < 0.05 and ## *p* < 0.01 compared to the normal group; * *p* < 0.05 compared to Aβ-treated group.

**Figure 4 nutrients-11-02591-f004:**
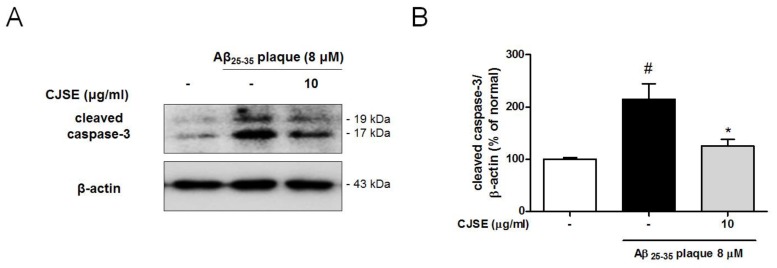
Effects of CJSE on expression of activated caspase-3 in mouse primary neuronal culture. (**A**) Representative band images of cleaved caspase-3 expression. (**B**) Quantifications of cleaved caspase-3 levels were normalized to β-actin. Values are indicated as the mean ± S.E.M. # *p* < 0.05 compared to the normal group; * *p* < 0.05 compared to Aβ-treated group.

**Figure 5 nutrients-11-02591-f005:**
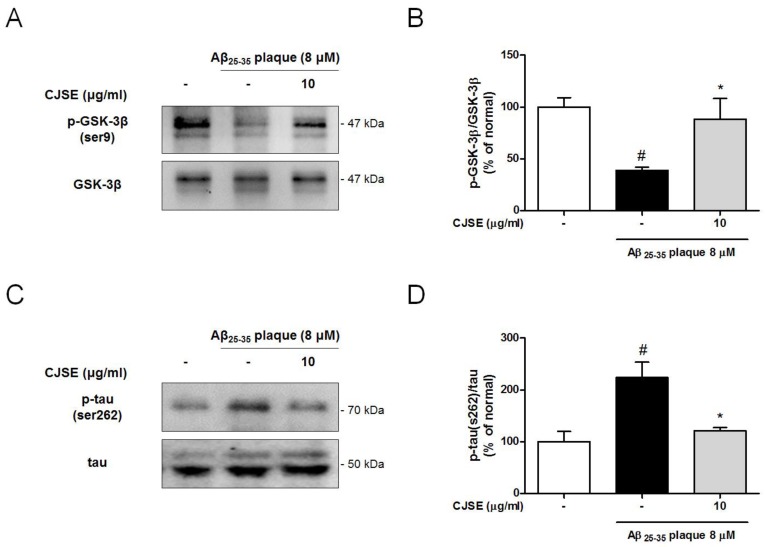
Effects of CJSE on the expression of phosphorylated GSK-3β and phosphorylated tau in mouse primary neuronal culture. (**A**,**C**) Representative band images of GSK-3β (**A**) and tau (**C**) expression. (**B**,**D**) Quantifications of the phosphorylated-form/total form ratios of GSK-3β (**B**) and tau (**D**). Values are indicated as the mean ± S.E.M. # *p* < 0.05 compared to the normal group; * *p* < 0.05 compared to Aβ-treated group.

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
