# Peer review of "Cuscutae Japonicae Semen Ameliorates Memory Dysfunction by Rescuing Synaptic Damage in Alzheimer’s Disease Models"

_nutrients, 2019, doi:10.3390/nu11112591_

Round 1

Reviewer 1 Report

The current study investigated the effects of Cuscutae Japonicae Semen (CJS), which is commonly used in traditional medicine in Asia, on Alzheimer’s pathology using mice and in vitro Alzheimer’s Disease models.

First, the authors treated 5xFAD transgenic mice from 20 to 28 weeks of age with CJS extract and administered the Morris water maze for 5 days during the last week of treatment in comparison to wild type (WT) or untreated 5xFAD transgenic mice. Mice were euthanized and their hippocampus (cornu ammonis and dentate gyrus regions) were evaluated for the protein levels of SYP and PSD-95. The authors found that the escape latencies of the untreated 5xFAD mice were significantly longer compared to WT mice on the last day of experiment and that escape latencies of the CJS treated 5xFAD mice were significantly shorter than untreated 5xFAD mice during the last two days of experiment. The authors also reported that while the protein levels of SYP and PSD-95 were significantly lower in the hippocampus of the untreated 5xFAD mice, compared to WT mice, the levels of both proteins were higher in CSJ treated 5xFAD mice compared to untreated mice. A similar effect was suggested by an in vitro model using mouse primary neuronal culture exposed to Ab plaque after CJS treatment or untreated conditions. The authors concluded that CJS treatment rescues synaptic loss induced by Ab synaptotoxicity.

Next, the authors evaluated the effect of CJS treatment on Caspase-3, as a potential mechanism by which CJS treatment may attenuates synapse degeneration induced by Ab toxicity. They used a primary mouse neuronal culture exposed to Ab plaque after CJS treatment or untreated conditions. The authors found that while Ab plaque elevated the levels of cleaved caspase-3 (which is associated to synaptic apoptosis) compared to Ab-unexposed neurons, the Ab-exposed neurons pre-treated with CJS had a lower increase of cleaved caspase-3.

Finally, Ab-exposed mouse primary neurons seem to have lower levels of p-GSK-3b and higher levels of p-tau compared to Ab-unexposed neurons, whereas CJS-treated neurons exposed to Ab plaque showed higher levels of p-GSK-3b and lower levels of p-tau compared to untreated neurons. Taken together, the authors suggest that CJS treatment can improve memory loss by rescuing caspase-3-mediated synaptic damage caused by Ab.

Overall, the research question is relevant, and the experimental design is scientifically sound. Systematic research on the safety, molecular effects, and clinical efficacy of the numerous herbs in common usage is needed and may enable medicinal-plant-based drug discovery. This research has potential to contribute with insights in this area. However, there are points to be revised or further discussed:

The methods part lacks important information how many mice were included per group, how many replicates were performed for the in vitro

How did the authors determine the CJS concentration to be administered to the mice and in vitro assays? The authors might have performed dose-response assays in a previous publication, but it is worth to mention the rational in the current manuscript. Moreover, by how much does the dose administered in mice in this study translates to human intake when regularly consumed as medicinal food?

If toxicity mediated by Aβ deposition is suggested to be reverted by CJS treatment by inhibiting the expression of cleaved caspase-3, I wonder if and what would be an CJS effect in a healthy mouse brain. It would be interesting to include a CJS-treated WT control group to evaluate potential effects of CJS on normal brain.

I would like to see the data that shows that CJS did not show any effects on Aβ accumulation in brain of CJS treated 5xFAD compared to the untreated 5xFAD mice, unfortunately, as stated in the discussion “data not shown”.

Overall, figures are quite small, and the pictures have poor resolution. For example, in Figure A – top panel with the pictures of CA3 it is impossible to see any differential protein expression across the conditions. Maybe the authors can find a more representative picture. How many replicates were used to generate the graphs in figure A?

The Western Blots are lacking a ladder or the band sizes. The manuscript lacks information on how many replicates were used for the protein quantification in the Western blots to generate the graphs. For instance, the blot shown in figure 5A does not look like it matches the data shown in the graph B.

Author Response

Dear Reviewer, we appreciate your kind comments and the opportunity to revise our work. We have responded to the comments and revised the manuscript after carefully considering your suggestions. 

Reviewer 2 Report

This manuscript is focused on evalulating the effects of Cuscutae Japonicae Semen (CJS) on memory loss as well as possible mechanisms of action. Specifically, the authors used AD transgenic mice and cell culture assays to address the synaptic damage as a possible mechanism of memory impairment. I think this paper warrants publication. There are a few points for the authors to address or consider. 

Please provide authentication of the plant, as it was purchased from local market.  There is a lot of key information relating to the animal experiment has been omitted eg power calculation to determine the group size , animal gender, route of CJSE administration.  The authors found synaptic changes in CA3 and DG (figure 2), what about CA1? It is valuable to include the Ab deposits data, although there is no statistical effect.  It is worthwhile to confirm the cell culture findings in vivo, eg caspase-3, GSK and tau. 

Author Response

(The authors gave the same response as above.)

Round 2

Reviewer 1 Report

Thank you for considering mine suggestions and have made the appropriate changes to the manuscript.